# The shape of cancer relapse: Topological data analysis predicts recurrence in paediatric acute lymphoblastic leukaemia

Salvador Chulián[1,2,3]*, Bernadette J. Stolz[4,5], Álvaro Martínez-Rubio[1,2,3], Cristina Blázquez Goñi[2,6,7], Juan F. Rodríguez Gutiérrez[2,6], Teresa Caballero Velázquez[7,8], Águeda Molinos Quintana[7,8], Manuel Ramírez Orellana[9], Ana Castillo Robleda[9], José Luis Fuster Soler[10], Alfredo Minguela Puras[11], María V. Martínez Sánchez[11], María Rosa[1,2,3], Víctor M. Pérez-García[3,12,13], Helen M. Byrne[4]*

**1** Department of Mathematics, Universidad de Cádiz, Puerto Real (Cádiz), Spain, **2** Biomedical Research and Innovation Institute of Cádiz (INiBICA), Hospital Universitario Puerta del Mar, Cádiz, Spain, **3** Department of Mathematics, Mathematical Oncology Laboratory (MOLAB), Universidad de Castilla-La Mancha, Ciudad Real, Spain, **4** Mathematical Institute, University of Oxford, Oxford, United Kingdom, **5** Laboratory for Topology and Neuroscience, École Polytechnique Fédérale de Lausanne, Lausanne, Switzerland, **6** Department of Paediatric Haematology and Oncology, Hospital Universitario de Jerez, Jerez de la Frontera (Cádiz), Spain, **7** Department of Haematology, Hospital Universitario Vírgen del Rocío, Instituto de Biomedicina de Sevilla (IBIS), Sevilla, Spain, **8** CSIC, University of Sevilla, Sevilla, Spain, **9** Department of Paediatric Haematology and Oncology, Hospital Infantil Universitario Niño Jesús - Instituto Investigación Sanitaria La Princesa, Madrid, Spain, **10** Department of Paediatric Haematology and Oncology, Hospital Clínico Universitario Virgen de la Arrixaca - Instituto Murciano de Investigación Biosanitaria (IMIB), Murcia, Spain, **11** Immunology Service, Hospital Clínico Universitario Virgen de la Arrixaca - Instituto Murciano de Investigación Biosanitaria (IMIB), Murcia, Spain, **12** Instituto de Matemática Aplicada a la Ciencia y la Ingeniería (IMACI), Universidad de Castilla-La Mancha, Ciudad Real, Spain, **13** ETSI Industriales, Universidad de Castilla-La Mancha, Ciudad Real, Spain

* salvador.chulian@uca.es (SC); helen.byrne@maths.ox.ac.uk (HMB)

**Data Availability Statement:** The authors confirm that all data underlying the findings are fully available without restriction. Implementation

## Abstract

Although children and adolescents with acute lymphoblastic leukaemia (ALL) have high survival rates, approximately 15-20% of patients relapse. Risk of relapse is routinely estimated at diagnosis by biological factors, including flow cytometry data. This high-dimensional data is typically manually assessed by projecting it onto a subset of biomarkers. Cell density and "empty spaces" in 2D projections of the data, i.e. regions devoid of cells, are then used for qualitative assessment. Here, we use topological data analysis (TDA), which quantifies shapes, including empty spaces, in data, to analyse pre-treatment ALL datasets with known patient outcomes. We combine these fully unsupervised analyses with Machine Learning (ML) to identify significant shape characteristics and demonstrate that they accurately predict risk of relapse, particularly for patients previously classified as 'low risk'. We independently confirm the predictive power of CD10, CD20, CD38, and CD45 as biomarkers for ALL diagnosis. Based on our analyses, we propose three increasingly detailed prognostic pipelines for analysing flow cytometry data from ALL patients depending on technical and technological availability: 1. Visual inspection of specific biological features in biparametric projections of the data; 2. Computation of quantitative topological descriptors of such projections; 3. A combined analysis, using TDA and ML, in the four-parameter space defined by

details, code and processed PH data are freely available on https://github.com/salvadorchulian/shapecancerrelapse. Anonymised, merged FC files are available in the flow respository http://flowrepository.org/id/FR-FCM-Z68U. All other data needed to evaluate the conclusions in the paper are present in the paper and/or the Supplementary Materials.

**Funding:** This work was partially supported by the Fundación Española para la Ciencia y la Tecnología (FECYT project PR214 to M.R.), the Asociación Pablo Ugarte (APU, Spain, to M.R.), Junta de Andalucía (Spain) group FQM-201 (to M.R.), Junta de Comunidades de Castilla-La Mancha (grant number SBPLY/21/180501/000145 to V.M.P.-G.), the Programme of Research and Transfer Promotion from the University of Cádiz (grant number EST2020-025 to S.C.), Ministry of Science and Technology, Spain (grant number PID2019-110895RB-I00 to V.M.P.-G.), Spanish National Plan for Scientific and Technical Research and Innovation (grant number PDC2022-133520-I00 to V.M.P.-G.). This work was also subsidized by a grant for the research and biomedical innovation in the health sciences within the framework of the Integrated Territorial Initiative (ITI) for the province of Cadiz (grant number ITI-0038-2019 to M.R., 80% co-financed by the funds of the FEDER Operational Program of Andalusia 2014-2020, European Regional Development Fund, Council of Health and Families). B.J.S. and H.M.B. are members of the Centre for Topological Data Analysis, funded by the EPSRC grant (EP/R018472/1). B.J.S. is further supported by the L'Oréal-UNESCO UK and Ireland For Women in Science Rising Talent Programme. S.C. was hired at the University of Cádiz with funding from the ITI project ITI-0038-2019. Á.M.-R. was hired at the University of Cádiz with funding from the APU project. The funders had no role in study design, data collection and analysis, decision to publish, or preparation of the manuscript.

**Competing interests:** The authors have declared that no competing interests exist.

CD10, CD20, CD38 and CD45. Our analyses readily extend to other haematological malignancies.

## Author summary

Acute lymphoblastic leukaemia (ALL) is a blood cancer which affects predominantly children and adolescents. Therapy typically fails in approximately 20% of patients, who suffer from relapse. To determine disease status, clinicians assess cell types, their interactions, as well as deviations from normal behaviour. Flow cytometry (FC) is a method that quantifies the intensity of specific cell markers, giving rise to high-dimensional data. This routinely collected information is then reduced to obtain human-interpretable visualisation for prognosis. Topological Data Analysis (TDA) is a field of mathematics that studies shapes in data, considering isolated data islands and empty spaces between them. We showcase how to use TDA to extract shape characteristics in FC data of relapsing patients. We propose three pipelines, of increasing methodological complexity, to aid clinical decisions for risk stratification in ALL. In combination with Machine Learning, TDA enables high-accuracy predictions of relapse to be made at the time of diagnosis.

## Introduction

Acute Lymphoblastic Leukaemia (ALL) is a haematological malignancy that affects all age groups, particularly children. Indeed, it is the most frequent type of paediatric cancer [1]. While current chemotherapy regimes have improved survival rates, as many as 15–20% of patients relapse following chemotherapy, i.e. the disease returns, even though the initial response to treatment may be good. Current protocols tailor the intensity of therapy to the risk of relapse, estimating this risk through a combination of biological information at diagnosis and the individual response to therapy. Patients at high risk of relapse receive more intensive chemotherapeutic regimes, before resistant clones emerge and become dominant [2, 3], improving survival rates. Very high risk patients are subjected to more aggressive therapies, such as bone marrow transplants or CAR-T-cell therapies [4]. Identifying patients at risk of relapse as early as possible and tailoring treatment plans accordingly is, therefore, of major clinical importance.

At diagnosis, clinicians typically collect a combination of quantitative biomolecular, and qualitative (usually binary) cell morphology and genetic data from a patient [5] to determine the type of haematological cancer, disease severity, i.e. prognosis, and likely response to treatment. Based on experience gained from multi-centre clinical trials, patient data is manually assessed to determine a patient's treatment plan, following an induction-consolidation-reinduction-maintenance chemotherapy regime, which is standard in many countries [6]. Many clinical trials also collect data about patient responses to treatment, including whether they experienced relapse. These data are ideally suited to retrospective analyses, not only to optimise treatment plans, but also to predict relapse. Exploiting existing datasets represents a promising approach to overcome many of the challenges in personalised medicine in oncology, such as patient phenotyping for prognosis or biomarker discovery [7–9].

Flow cytometry (FC) data are routinely used for the diagnosis of haematological malignancies such as ALL [10]. These data provide quantitative information about the bone marrow status at the level of single cells. A particular challenge of FC data is its high-dimensionality: each

cell from a patient sample is represented by one data point whose coordinates correspond to the intensities of a set of predetermined fluorescent biomarkers. This collection of data points is also referred to as a point cloud. The biomarkers usually comprise immunophenotypic (IPT) markers [11] such as CD19, CD10 and CD20, which are characteristic of B-lymphocytes subpopulations, more general markers such as CD45, CD38 and CD34, which indicate cell differentiation and maturation status, as well as CD3, CD33 or cyMPO, which are specific to other cell lineages, e.g. T-lymphocytes or myeloid cells. The current strategy for managing the high-dimensionality in these data is to generate two-dimensional projections of pairwise combinations of biomarkers [12], and then to manually assess the shapes of the projected data. For example, the shapes can be inspected to identify subsets of points within the point cloud that correspond to cancerous cells. Indeed, previous studies have highlighted the importance of 'empty spaces' in FC data over the course of treatment [13–15].

However, manual assessment is time-consuming, subjective, and prone to errors. Therefore, more systematic and automated methods of analysis are required. In the context of FC data, Machine Learning (ML) classifiers have been used to predict relapse in B-ALL patients, yet these studies focus on either minimal residual disease follow-up, i.e., the number of cancer cells after treatment [16], or on biomarker intensity levels in different B-ALL subpopulations [17]. Several methods have been developed for application in other diseases [18], mostly concentrating on dimensionality reduction and cell population identification. Others, such as CellCNN [19], CITRUS [20], and FloReMi [21] employ convolutional neural networks (CNN), LASSO, and Random Forest models, respectively, to perform classification. All these approaches, however, have one thing in common: they rely on the level of expression of cell markers, i.e. their fluorescence intensity values, for classification and feature extraction. The clinical approach, however, focusses on a higher-level description of the data: the shape, orientation and structure of the point cloud. Therefore, we propose the integration of topological biomarkers which characterise the shape of the data with Machine Learning methods. In this way, we can exploit the advantages of each approach to improve the value of these techniques in biomedical data analysis.

Topological data analysis (TDA) [22, 23] is a set of methods that uses ideas from the mathematical discipline topology to study the 'shape' of high-dimensional data. The most prominent method in TDA is persistent homology (PH) [22–26]. PH quantifies features, such as empty spaces, in point cloud data, at different spatial scales. Thus, in contrast to other data analysis methods, it is not necessary to predetermine a spatial scale at which relationships between data points are considered; PH takes into account the full range of possible scales. PH describes the empty spaces of data via topological features such as connected components (i.e. clusters of data points), and loops (i.e. data points surrounding an empty hole). The output from PH are barcodes which correspond to topological features in the data. Features that appear across a range of scales in these barcodes, are called 'persistent' and are often viewed as particularly characteristic of a dataset (for an example, see Fig 1E and 1F). The mathematical foundations of PH ensure that small changes in the data lead to only small changes in the barcodes. PH is readily computable [26], robust to noise [27] and its outputs are interpretable. In recent years, improved computational feasibility of PH has increased its applications to (high-dimensional) data in many contexts [28, 29], including studies of the shape of brain arteries [30], neurons [31], the neural code [32], airways [33], stenosis [34], zebrafish patterns [35], ion aggregation [36], contagion dynamics [37], proteins [38], spatial networks [39–43], and geometric anomalies [44]. In oncology, PH has been used to construct new biomarkers [40, 45, 46], to classify tumours [47, 48] and genetic alterations [49] and to quantify patterns of immune cell infiltration into solid tumours [50].

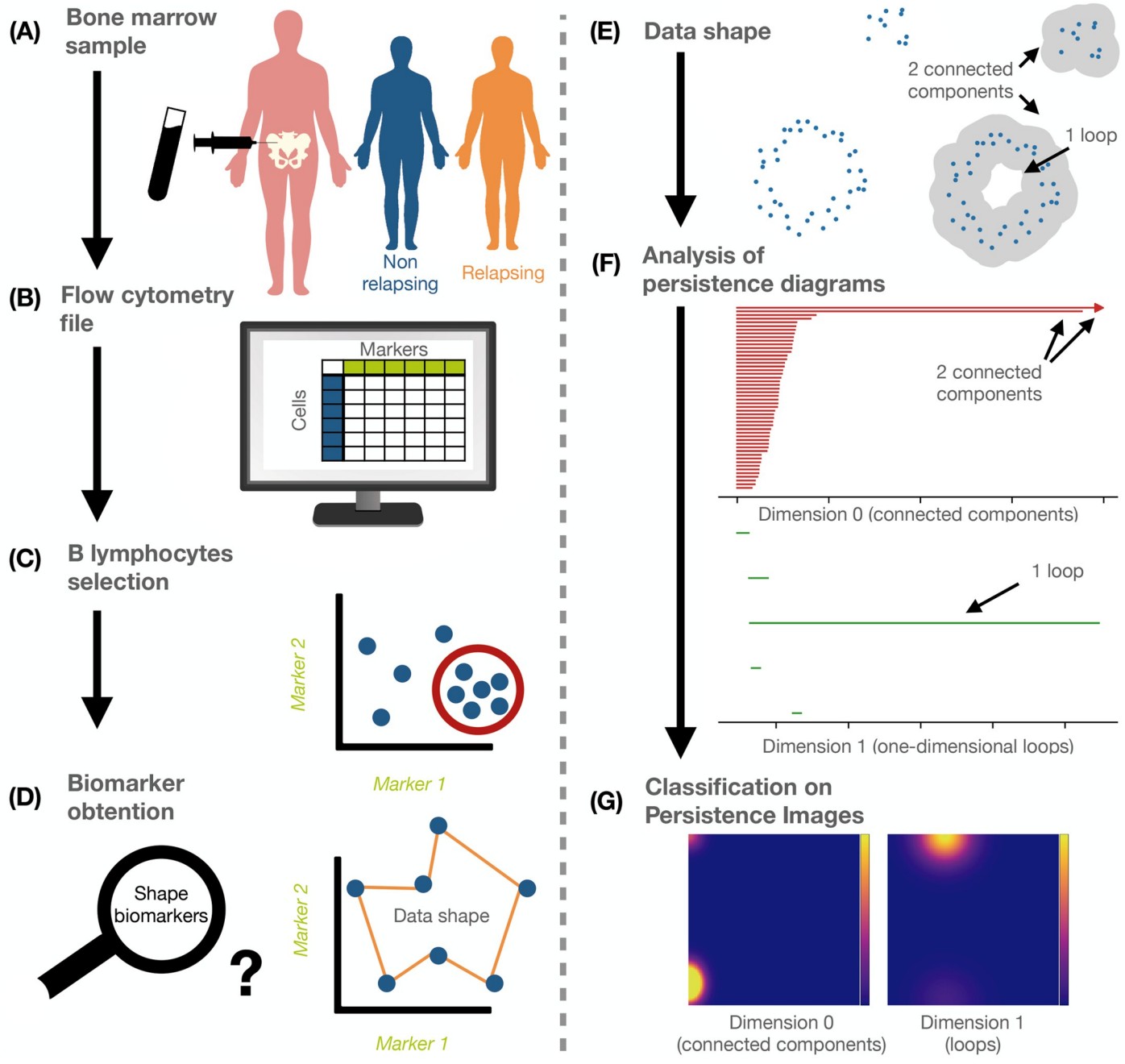

**Fig 1. Flowchart of methodology. (A)** We collected bone marrow data from relapsing (R) and non-relapsing (NR) B-ALL patients at the time of diagnosis. **(B)** These samples were analysed via flow cytometry (FC) which gave rise to single-cell data with information on immunophenotypic (IPT) markers. **(C)** We pre-processed these data and selected the patients' B-lymphocytes. **(D)** Our aim was to obtain quantitative biomarkers from the IPT markers that can be used to identify patients at risk of relapse. **(E)** We conducted topological data analysis (TDA), which quantifies the shape of data via topological characteristics. **(F)** An important tool in TDA is persistent homology (PH), which can be used to obtain summaries in the form of persistence barcodes. Barcodes are topological signatures of the data and show topological features such as connected components and loops as well as their scale in the data. **(G)** Persistence barcodes can be transformed into vector summaries called Persistence Images. As a second approach, we used these images for classification.

In this paper we design a novel pipeline to analyse FC data that combines PH and ML. TDA has been successfully applied to study COVID-19 in FC data [51]. We apply our pipeline to data collected at three different hospitals in Spain [52] from patients with the most common subtype of paediatric ALL [53], one that is characterised by rapid growth of early

B-lymphocytes, B-ALL. We demonstrate that our shape-based analysis enables us to predict relapse in ALL patients with very high accuracy (see Fig 1) and we independently confirm the known biomarkers CD10, CD20, CD38 and/or CD45 as most relevant for relapse (R) versus non-relapse (NR) classification. We further identify specific topological features (connected components and loops) which characterise patients at risk of relapse in 2D projections. Finally, we highlight that by including vectorisation methods (Persistence Images [54]) our pipeline can make full use of the topological information available in a 4D projection of the FC data, and outperform the commonly used 2D projections in detecting patients at risk of relapse. Our approach can be readily generalised to many other haematological malignancies. Additionally, TDA provides an interpretative layer to ML techniques, which clinicians have traditionally identified as a bottleneck [55]. Further, this contributes to the recent advances in interpretable ML [56].

Based on our findings, we suggest three novel approaches of increasing complexity for assessment of risk of relapse in B-ALL. The first approach is derived from our insights from PH and suggests shape characteristics that are specific to 2D projections of data from patients at risk of relapse which can be included in visual inspection when PH is not available. If PH is available for data analysis but not ML, we suggest a second approach where barcodes are summarised by computing how many connected components (topological features in dimensions 0) and loops (topological features in dimension 1) persist longer than a given range of thresholds. The third approach includes a combination of PH and ML and is fully automated. It achieves optimal accuracy in relapse prediction compared to manual assessments performed at diagnosis, and also in patients who with conventional methods were falsely predicted to have low risk for relapse. To our knowledge, this is the first study that quantifies (rather than qualitatively assesses) 'empty spaces' in FC data to predict relapse patients at the time of diagnosis.

## Results

### Machine Learning classifier applied to TDA independently confirmed state-of-the-art markers for prognosis

We applied ML and TDA descriptors of patient flow cytometry (FC) data. With our quantitative TDA descriptors, we independently confirmed the prognostic relevance of empty spaces in data projections of state-of-the-art biomarkers. The general pipeline used is shown in Fig 1, where we classified patients into either the non-relapsing (NR) group, i.e. patients who show no refractory values for minimal residual disease after three years, or the relapsing (R) group. Our data contained 80 NR patients and 16 R patients. We divided the datasets into discovery and validation sets, as described in 'Methods' and Table A in S1 Text.

In line with standard clinical practice to assess prognosis, we constructed 2D projections of the high-dimensional point clouds, considering all pairwise combinations of the 16 immunophenotypic (IPT) markers for each patient (see Fig 2A–2C). We then applied PH (specifically, a Vietoris-Rips filtration, see 'Methods') and obtained PH barcodes for dimension 0, i.e. connected components, and dimension 1, i.e. loops (see Fig 2D; and, for a detailed description, see 'Methods'). In a first step, we extracted simple descriptors from each barcode to create vectors for each patient, for each of the pairwise combinations of markers (see Fig 2E).

We then performed a Random Forest analysis, with cross-validation, on each patient subgroup within the discovery set, assigning 60% of the patients to a training group, and 40% to a testing group. After classification, we excluded those IPT pairs with low accuracy (see 'Methods'), where the IPT markers that appeared most frequently in the remaining pairs were

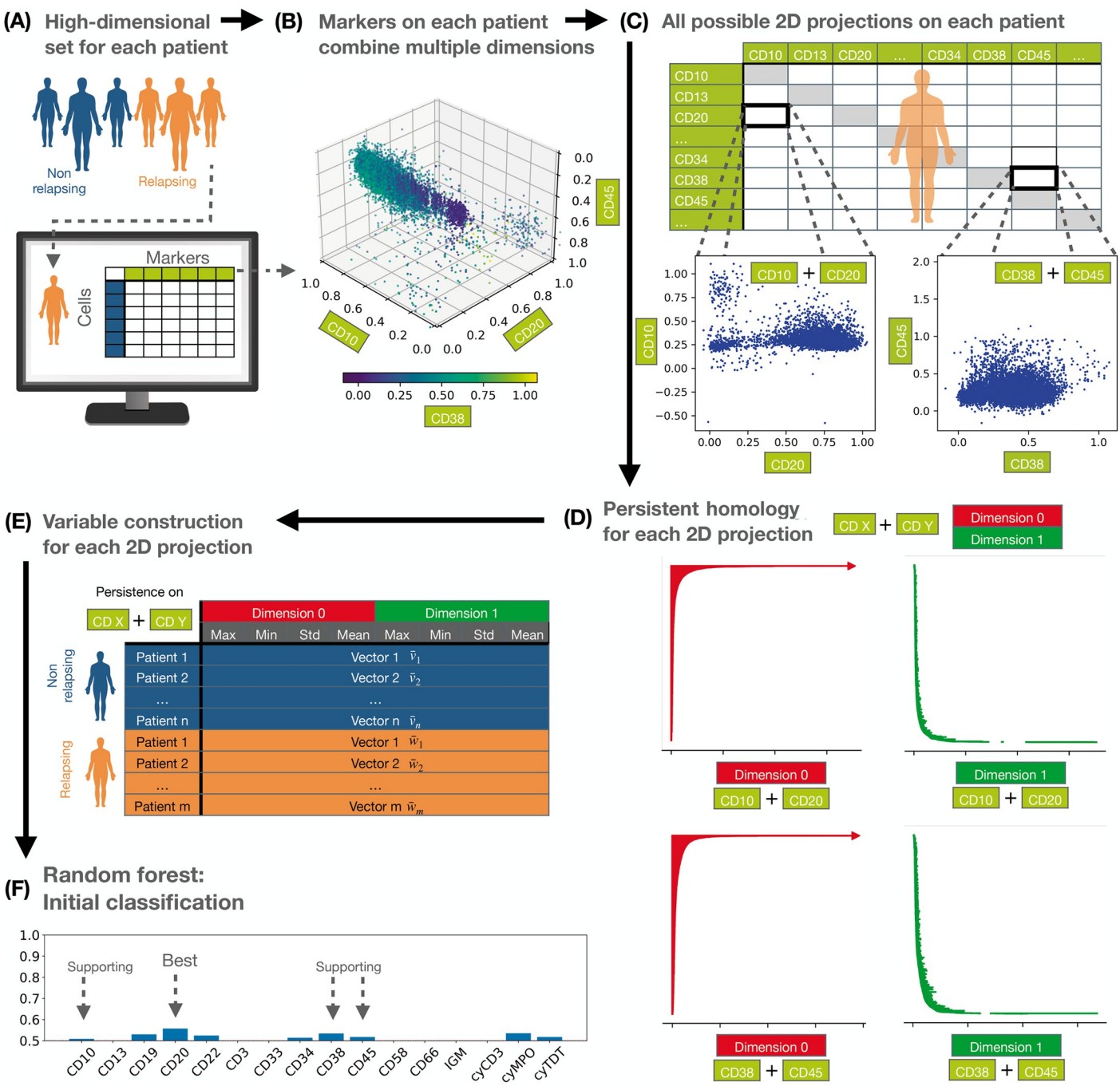

**Fig 2. Workflow for confirming prognostic, state-of-the-art markers.** (**A**) We collected data from non-relapsing (NR) and relapsing (R) patients containing 16 immunophenotypic (IPT) markers. (**B**) To reduce the dimensionality of the data and enable TDA computations, we first selected a subset of markers that contribute most to differences in the shape of the data between the R and NR groups. (**C**) We constructed pairwise projections of the selected IPT markers in line with standard clinical practice. (**D**) We applied PH to each patient point cloud, and quantified connected components and loops in the data. (**E**) For each pairwise combination of IPT markers associated with a particular patient, we computed an 8-dimensional feature vector summarising information from the PH analysis. (**F**) We applied Random Forest classification to the feature vectors and confirmed that the biomarkers CD10, CD20, CD8 and CD45 contain information that is most relevant for distinguishing R from NR patients based on the shape of their data.

CD10, CD19, CD20, CD22, CD34, CD38, cyMPO and cyTdT (see Fig 2F and S1 Data). Only combinations including CD10 or CD20 had a mean accuracy over 70%.

Next, we considered the validation dataset to verify that the selected markers provided relevant shape information. The set of pairwise combinations yielded an additional 5520 persistence barcodes in each dimension. As before, we studied these barcodes using Random Forest analysis, with k-fold cross-validation to check for robustness. When we used oversampling, our results showed no significant differences for different biomarkers (see S2 Data). However, when we used the unbalanced data and stratified k-fold, i.e., maintaining the proportion of samples for each class, only IPT markers CD10, CD20, CD38 and CD45 produced mean AUCs>50% (see S1 and S3 Data). To establish whether these markers carried independent information, we also computed the correlations between the markers. None of the IPT markers showed strong correlations in either the NR or R groups (see Figs F and G in S1 Text). Biologically, it is well-known that CD10, CD20 and CD45 identify subpopulations within the CD19$^+$ B-lymphocyte clouds [57], while CD38 is a marker of aberrant B-lymphoblasts [58, 59]. While the expression levels of these biomarkers have been used to predict B-ALL relapse [17], empty spaces in the data have not previously been quantified. Our independent Random Forest analysis, performed on simple topological descriptors, confirmed the importance of these biomarkers. We performed our subsequent analysis on a reduced dataset containing only these variables.

## PH identifies connected components and loops characteristic of R patients

Following our selection of the four most informative biomarkers for prognosis (CD10, CD20, CD38 and CD45), our goal was to use PH to compute differences in the number and persistence of connected components and loops in the data point clouds from R and NR patients. First, we applied PH to all pairwise combinations of these biomarkers for each patient. We visualised the persistence of connected components (dimension 0) and loops (dimension 1) by constructing what we term *persistence threshold (PT) curves*; PT curves reveal characteristic topological features, i.e. persistent features, for each patient cohort (see 'Methods'). We averaged the PT curves for both patient cohorts and all pairwise combinations of biomarkers and found significant differences between the discovery and validation sets. The results for the mean PT curves for all pairwise combinations are presented in Figs J-O in S1 Text for both dimensions 0 and 1, i.e. connected components and loops.

We present illustrative results for CD10-CD20 in Fig 3. In both the validation and discovery datasets, the mean values of the dimension 1 PT curves were larger in R patients than in NR patients (see Fig 3A.2). These differences were statistically significant in dimension 1, where $\tau$ is the persistence threshold with $\tau \in [0.04, 0.05]$ (see 'Methods' for other threshold ranges). An illustrative example for dimension 0 for CD10-CD20 is presented in Fig I in S1 Text.

Intuitively, there are fewer persistent loops, i.e. empty spaces, in the NR patients than in the R patients as the loops in the NR patients tend to be 'narrower' and disappear at smaller spatial scales (radius $r = 0.03$ versus $r = 0.06$, see Fig 3B.1 and 3B.2, respectively). In the barcodes, less persistent loops correspond to shorter bars. This is captured by the PT curves. In dimension 0, we observed the same trends for CD10-CD20: the number of less persistent connected components in the data of R patients was larger than that of NR patients (see S1 Text for details). Complete results for all pairwise combinations are included in Figs N and O in S1 Text, respectively; they show the absolute and relative numbers of bars that are larger than a fixed threshold value for each patient. The number of persistent loops was higher in R patients for all combinations that included either CD10 or CD20, except for the combination CD10-CD45. We only found a higher number of persistent loops for NR patients in CD38-CD45. The

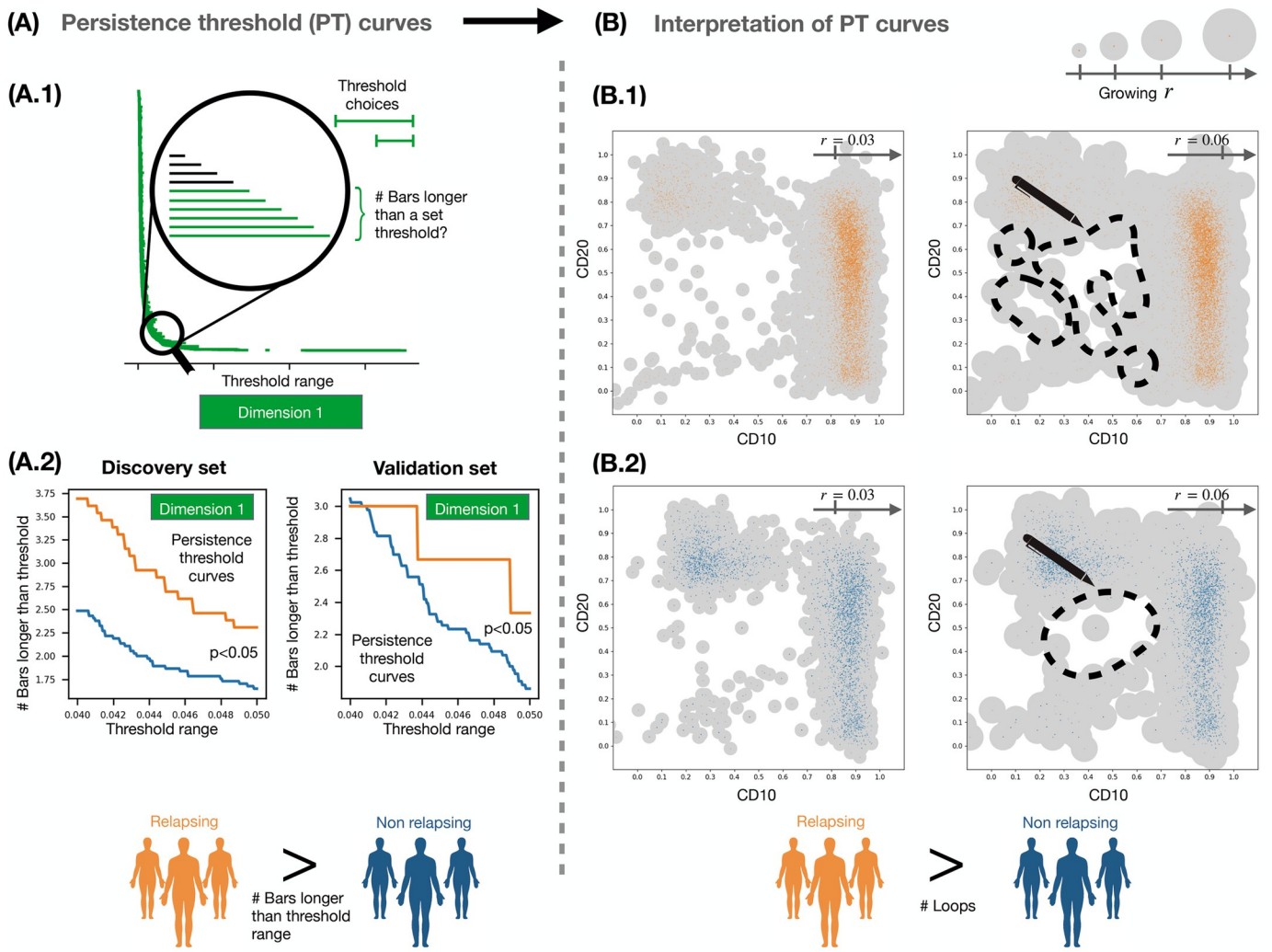

**Fig 3. Analysis and interpretation of output from persistent homology. (A)** We computed the number of bars in dimension 1 barcodes whose lengths exceed a set of threshold values (long persistence for $\tau \in [0.04, 0.05]$, medium persistence for $\tau \in [0.007, 0.015]$). The resulting persistence threshold curves provide information about the persistence of loops in the data. In both the discovery and validation sets, the number of bars for all pairwise combinations of biomarkers CD10, CD20, CD38 and CD45 differ statistically significantly for R and NR patients, except for the case of CD10-CD45. In this case, the persistence threshold curves for R patients contained a higher number of persistent loops than for NR patients. **(B)** Based on our quantitative analysis, we suggest a qualitative approach for analysing the 2D projection CD10-CD20 (in cases where a full quantitative approach is not possible): consider balls centred at each data point whose radii increase; our results indicate that, as the ball radii grow, individual loops merge more quickly for NR patients (B.2) than for R patients (B.1).

number of non-persistent loops was only larger for NR patients than for R patients for the combination CD20-CD38. In dimension 0, connected components with short persistence were only found in R patients for the combination CD10-CD20. These results were statistically significant and consistent between discovery and validation datasets.

When we applied the same analysis to the 4-dimensional point clouds of each patient, the differences between the R and NR group were not significant in the whole dataset (see S1 Text). As before, we first performed statistical analyses on simple barcode descriptors, i.e. the maximum, minimum and mean persistence and the standard deviation, extracted from the dimension 0 and 1 barcodes of the NR and R groups. Since the resulting p-values for all variables were larger than 0.05, we concluded that, in general, the simple barcode descriptors are unable to discriminate between the two groups (see results summary in Fig H(A) in S1 Text).

Given the analogy between clusters and connected components, we performed clustering in the data point cloud to verify the dimension 0 PH results, and thus looked for statistical differences between R and NR patients. We considered 4-dimensional patient point clouds in the discovery set and applied the FlowSom algorithm [60] (see 'Methods' and Figs P-U in S1 Text) to each patient point cloud to determine the number of clusters. We then compared results between both cohorts. The number of clusters in point clouds from NR patients (4.01 ± 1.17) was lower (t-test, p-value 0.09) than the number in point clouds from R patients (4.56 ± 1.11). When we performed the same analysis with the 2-dimensional data, the same trends were observed for the 2D projections of CD10-CD20 (t-test, p-value 0.18), with fewer clusters in NR patients (3.51 ± 0.73) than in R patients (3.81 ± 1.17). While these results were marginally significant, they are consistent with those obtained from the persistence threshold curves (see Fig 3).

## Persistence Images accurately identified relapsing patients making full use of 4D data

In our previous analyses, we computed simple barcode descriptors from the PH barcodes which are summaries and, thus, inevitably lose some of the information. To fully exploit the information on empty spaces in the 4-dimensional datasets (with biomarkers CD10-CD20-CD38-CD45; see Fig 4A), we transformed the barcodes into Persistence Images (PIs) [54]. This method transforms barcodes into matrices which can be vectorised and, as such, are more amenable to analysis while also being stable to noise and maintaining an interpretable relationship to the barcodes from which they were generated (see Fig 4B and Section S1.4 and S2.3.c-d for methodological details in S1 Text).

We used PIs in matrix form as input to Logistic Regression (LR) and Support Vector Machine (SVM) analyses. We considered three variables: datasets studied (balanced, or not, by oversampling), dimensions of the topological features analysed (0, 1 or 2) and biomarkers included (CD10, CD20, CD38, CD45). We included dimension 2 barcodes in this analysis by reducing the number of data points from $10^4$ to $10^3$.

We used the discovery and validation datasets for training and testing, respectively. The classification results are presented in Table 1. We performed cross-validation within the training sets to obtain representative SVM parameters as described in 'Methods'. We obtained perfect classification scores and 100% accuracy. By comparison, a Random Forest analysis of clinical variables which are routinely used for relapse risk assessment (such as age, sex and other molecular biology data as presence of mutations) lead to an accuracy of 86% after cross-validation, and an expert clinical analysis based on the same data lead to an accuracy of approximately 91% (for more details, see Section 2.1.c in the S1 Text). These differences depend on the hyperparameters used to construct the Persistence Images (PIs), and were obtained for the discovery and validation datasets (see Fig W in S1 Text for details). We constructed the mean PIs for R and NR patients to visually compare such differences between the two cohorts. Additionally, we computed the coefficients of the decision function in the LR classification and visualised the discriminatory areas between both cohorts in an image. Moreover, since R patients constituted approximately 20% of the total number of patients, we performed oversampling (see 'Methods') and reran our analysis with SVM. As before, we obtained 100% accuracy, with better classification for dimensions 0 and 1 than for dimension 2. We obtained the same trends when all three PIs (in dimensions 0,1 and 2) were used together as input for the classification methods (see Table D in S1 Text). Thus, the PH methods combined with the ML techniques were able to identify the R patients from the diagnostic FC data with high accuracy.

We also performed classification for a subgroup of patients, specifically those deemed to be of low or intermediate risk at time of diagnosis. From a clinical perspective, it would be

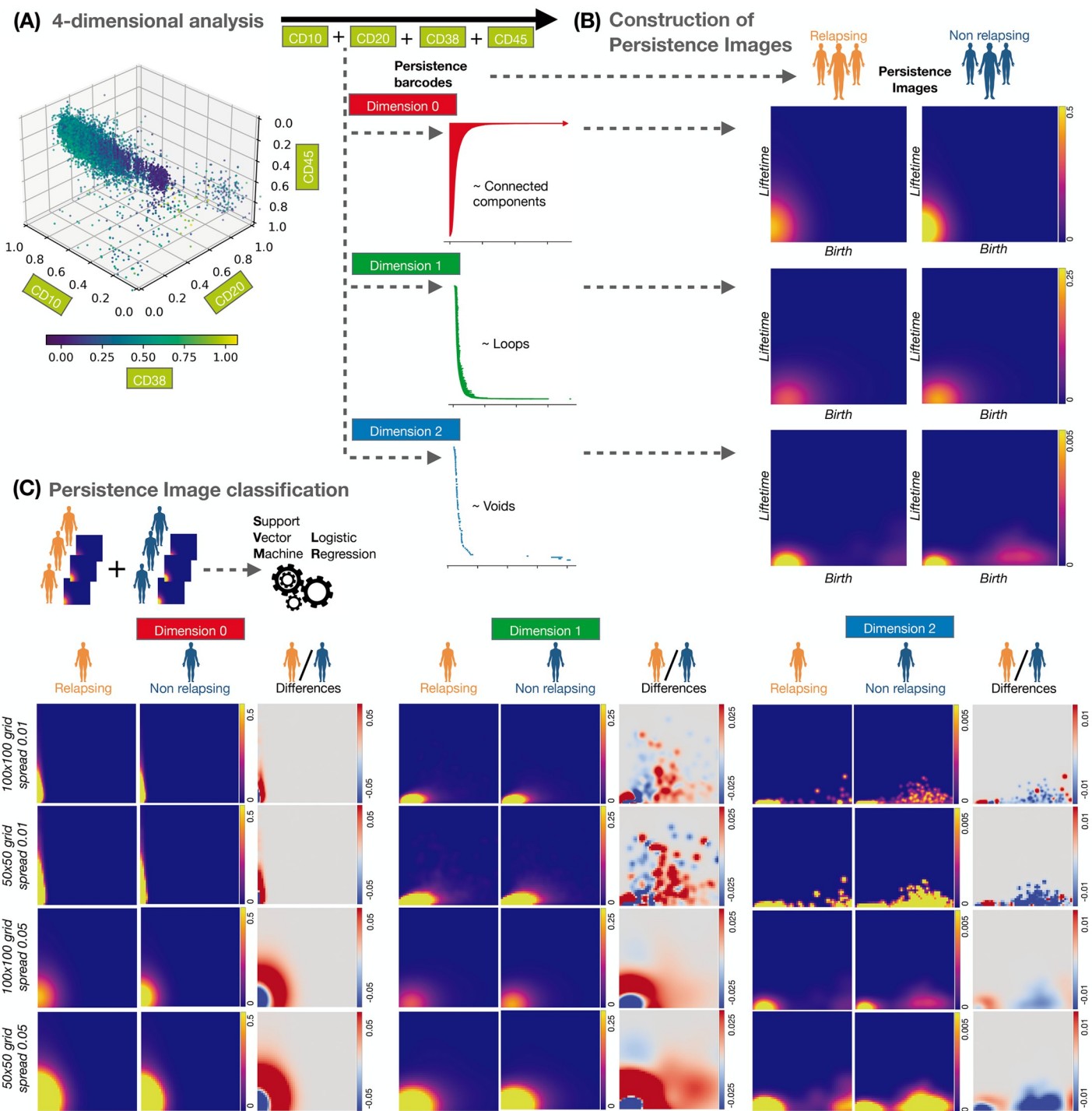

**Fig 4. Analysis using Persistence Images (PIs). (A)** We computed PH barcodes in dimensions 0 (connected components), 1 (loops) and 2 (voids in 3D) on the 4-dimensional space of biomarkers CD10-CD20-CD38-CD45. **(B)** We transformed the barcodes for each dimension into PIs, i.e. a matrix of intensity values. This transformation depends on hyperparameters, such as the choice of grid size and the spread of the Gaussian distribution which is used to convert barcode information into an image (see 'Methods' for a precise description). **(C)** The PIs (one for each patient in each dimension) were the input for classification using SVM and LR. We present the mean images for each patient cohort, as well as their differences.

**Table 1. Summary of classification scores of Persistence Images (PIs) after cross-validation on the discovery and validation datasets.** We constructed classifiers using logistic Regression (LR) and Support Vector Machines (SVM) on the discovery and validation datasets, using PIs obtained from PH analysis of biomarkers CD10, CD20, CD38, and CD45. The accuracy of the SVM method depends on the dimension of the barcode, the spread of the Gaussian and grid size used to generate the PIs, as well as the parameters $C$ and $\gamma$ for the SVM method. The accuracy (Acc.) of the model predictions after 6-fold cross-validation is greatest for Dimension 0 and 1. Additional summaries are provided in Tables C-E in S1 Text. An expanded analysis, using fixed hyperparameters and other classification scores (AUC, mean accuracy, standard deviation of the accuracy, recall, precision, F1-score and confusion matrices) can be found in Tables F and G in S1 Text.

|  | Spread | Grid | Method | Dim. 0 | Dim. 1 | Dim. 2 |
|---|---|---|---|---|---|---|
|  |  |  |  | Acc. | Acc. | Acc. |
| **Without oversampling** | **0.05** | **100x100** | LR | 0.93 | 0.93 | 0.93 |
|  |  |  | SVM | 1 | 1 | 0.94 |
|  |  | **50x50** | LR | 0.97 | 0.93 | 0.93 |
|  |  |  | SVM | 1 | 1 | 0.94 |
|  |  | **25x25** | LR | 1 | 0.93 | 0.93 |
|  |  |  | SVM | 1 | 1 | 0.95 |
|  |  | **10x10** | LR | 1 | 0.95 | 0.93 |
|  |  |  | SVM | 1 | 1 | 0.91 |
|  |  | **5x5** | LR | 1 | 0.95 | 0.93 |
|  |  |  | SVM | 1 | 1 | 0.82 |
| **With oversampling** | **0.05** | **100x100** | SVM | 1 | 1 | 0.94 |
|  |  | **50x50** | SVM | 1 | 1 | 0.95 |
|  |  | **25x25** | SVM | 1 | 1 | 0.96 |
|  |  | **10x10** | SVM | 1 | 1 | 0.95 |
|  |  | **5x5** | SVM | 1 | 1 | 0.87 |

advantageous to identify those patients whose initial risk assessment was incorrect and then to reclassify them. As shown in Table B in S1 Text, it is noteworthy that 85.7% of all patients who experience relapse were initially categorized as having a low-intermediate risk profile. We repeated our TDA and ML based classification of R versus NR. We classified the full dataset with SVM, with 20% training test and 6-fold cross-validation. The results are included in Table 2, where patients with low-intermediate risk were classified with high accuracy using the dimension 0 or 1 information from the PIs, both with and without oversampling.

**Table 2. Summary of classification scores of Persistence Images (PIs) for patients with low-intermediate risk.** An SVM classifier was constructed for patients with low or intermediate risk at the time of diagnosis, using PIs obtained from PH analysis of biomarkers CD10, CD20, CD38, and CD45. These results depend on the dimension of the barcode, the spread of the Gaussian and grid size used for the PIs. We fixed parameters $C = 10$ and $\gamma = 0.001$ for the SVM method, based on the previous results. The accuracy (Acc.) of the model predictions after 6-fold cross-validation is greatest for Dimensions 0 and 1. An expanded analysis with additional classification scores (AUC, mean accuracy, standard deviation of the accuracy, recall, precision, F1-score and confusion matrices) can be found in Tables H-I in S1 Text.

|  | Spread | Grid | Method | Dim. 0 | Dim. 1 | Dim. 2 |
|---|---|---|---|---|---|---|
|  |  |  |  | Acc. | Acc. | Acc. |
| **Without oversampling** | **0.05** | **100x100** | SVM | 1 | 1 | 0.94 |
|  |  | **50x50** | SVM | 1 | 1 | 0.94 |
|  |  | **25x25** | SVM | 1 | 1 | 0.93 |
|  |  | **10x10** | SVM | 1 | 1 | 0.89 |
|  |  | **5x5** | SVM | 1 | 1 | 0.84 |
| **With oversampling** | **0.05** | **100x100** | SVM | 1 | 1 | 0.93 |
|  |  | **50x50** | SVM | 1 | 1 | 0.94 |
|  |  | **25x25** | SVM | 1 | 1 | 0.95 |
|  |  | **10x10** | SVM | 1 | 1 | 0.86 |
|  |  | **5x5** | SVM | 1 | 1 | 0.89 |

## Discussion and conclusions

Predicting risk of relapse after treatment is one of the major challenges in cancer, particularly now that effective second and third-line treatments are available. In B acute lymphoblastic leukaemia (B-ALL), first-line chemotherapy regimes have increased overall survival by up to 90%, but a substantial fraction of patients relapse. Being able to start other treatments as soon as possible, i.e. before resistant clones emerge and become dominant [2, 3], could significantly improve treatment outcomes in these patients. In the present work we focused on predicting risk of relapse based on FC data, which is routinely collected at diagnosis of haematological malignancies. Clinicians manually assess the shape of 2D projections of this high-dimensional data for diagnosis and visualise its empty spaces. We combined PH, a method from TDA, with ML to obtain quantitative descriptors which are directly related to empty spaces which have been qualitatively observed in FC data. Our descriptors distinguish R and NR B-ALL patients at diagnosis with high accuracy. We focused our analysis around the following topological features: connected components (dimension 0) and loops (dimension 1).

We first used PH together with ML to detect markers relevant for the classification of R patients in an unsupervised manner. IPT markers CD10, CD20, CD45 and CD38 were among those with the highest relevance for classification. These biomarkers are characteristic of the B-cell line; CD10, CD20 and CD45 are routinely used to identify early B cell sub-populations [57], while CD38 is known to have prognostic value for B-ALL [52, 59]. Thus, our results confirmed the importance of these four biomarkers and, therefore, we retained only these four biomarkers for further analyses.

We analysed the 4D dataset (with the above biomarkers) and pairwise, 2D projections of the data with PH and identified those features which most influence the shape of the point clouds. Interestingly, using simple barcode descriptors extracted from the 2D data we were able to more accurately distinguish between R and NR patients than when considering the 4D data. Our ability to distinguish between R and NR patients increased further when barcodes from the 4D data were transformed into PIs, i.e. when more of the topological information provided by PH was used for classification. This implies that while the shape of the 2D projections can give insights via both visual inspection and crude shape descriptors, PH captures valuable information that goes beyond these features in the 4D data. This additional information can only be exploited when the data is transformed into a format that can be combined with ML while retaining as much information from the barcodes as possible.

In the 2D data, by considering differences between the R and NR PT curves, we found that R patients were characterised by larger numbers of persistent connected components and loops (see Fig 3B). In terms of biological insight, these results are consistent with the bone marrow cell population of R patients being more phenotypically diverse than those of NR patients. In the R patients, such cellular heterogeneity can be linked to non-dominant leukaemic sub-populations that might be drivers of disease relapse [61]. Increased cellular heterogeneity in R patients manifests as a higher number of isolated leukaemic point clouds (i.e. clusters of cells spread across phenotype space) in their projected FC data. In particular, the projections of marker combinations, including CD10 and CD20, show such B-cell subpopulations in the data. This may also explain why there are more loops in the data of R patients in the corresponding marker combinations; a more heterogeneous cell population leads to more and larger voids in the data which can result in more persistent loops. This is also related to the presence or absence of 'empty spaces' in FC data: in NR patients there is a lower number of 'empty' locations, where possible future cancer cells can reappear. However, the presence of cancer cells in such locations create more loops in the R patients dataset. In contrast, for NR patients we found more persistent loops in the marker combination CD38-CD45, the only

combination not including either CD10 or CD20. A possible explanation is that long-time-survival patients with B-ALL express higher levels of CD38$^+$ in their FC data, which could lead to homogeneity in NR patients. These considerations may also explain why 4D analysis of PT curves did not discriminate well between R and NR groups: we observe more isolated clusters and heterogeneity in the data point clouds of R patients in CD10-CD20 projections, while the intensity of CD38$^-$ in CD38-CD45 projections (which could be related to more homogeneity in these biomarkers for R patients) points to bad prognosis [52, 58]. In general, these significant differences do not seem to be associated with a consistent threshold range in either dimension, although dimension 1 features tend to be present at lower thresholds. As such, we recommend repeating the parameter scan when analysing new data.

In the 4D data, PIs which use the full information content of the barcodes rather than summaries (simple barcode descriptors and PT curves) enabled us to classify the data from R and NR patients via ML. We computed PIs from the barcodes in dimensions 0, 1 and also 2 (voids) for this analysis. The construction and classification of the PIs depended on the choice of several hyperparameters, such as the grid size, the spread of the Gaussian distribution, and the SVM classification parameters $C$ and $\gamma$ (see Fig W in S1 Text and 'Methods'); their values were optimized for each case. We note that we were able to perform high-grade classification using a small number of pixels (PI grid size of 5x5) and with fixed SVM hyperparameters $\gamma = 10^{-3}$, $C = 10$. Likewise, the classification scores were higher with a Gaussian spread of 0.05. Following balancing by oversampling, the classification was excellent in every dimension. On average, classification scores were higher for dimension 0 than dimension 1, and lower for dimension 2. Concatenating PIs from all three dimensions yielded the same trends as those for dimensions 0 and 1. Regardless of the classification method, dimension 0 attained a 100% classification score. Similarly, for dimension 1, the results showed a high score for almost every choice of hyperparameters in the construction of the PIs. We obtained slightly worse results in dimension 2 when Logistic Regression was used for classification. However, we note that, due to the high computational costs of PH in dimension 2, only a subset of $10^3$ data points were used, instead of the full number of $10^4$ points used to construct PIs in dimensions 0 and 1. SVM analysis applied to the PIs from dimensions 0 and 1 achieved 93% and 100% accuracy in the original and balanced data, respectively. Even when a 5x5 grid was used to generate the PIs the accuracy of SVM was maintained. This contrasted with existing classifiers, which have an accuracy (area under the curve) of less than 92%, with a similar sample size [17]. A previous analysis of Datasets 1 and 2 [52] using percentile vector curves computed from the distribution of the immunophenotypic intensity only lead to a classification accuracy of 80% between R and NR patients. In our dataset used here, we also computed the number of clusters in 2D projections of the data using markers CD10-CD20. Such analysis is related to dimension 0 persistent homology which considers connected components. However, the number of clusters in the projection of the data on CD10-CD20 only resulted in marginally significant results, with a $p$-value of 0.18. By considering the current protocols for risk stratification, 85.7% of patients that eventually relapsed were assessed at the time of diagnosis as having low or intermediate risk (see Table B in S1 Text and S4 Data). The results in Table 2 show how our method improves this initial classification at diagnosis compared to clinical variables and other molecular biology data (for more details, see Section 2.1.c in the S1 Text, and the previous molecular biology correlations for this dataset in Ref. [52]).

Our results show how using PH to analyse routine high-dimensional FC data can significantly improve the accuracy of predictions of relapse for patients with B-ALL. Our levels of accuracy and precision highlight the potential for topological methods to dissect phenotypic characteristics of B-ALL relapse. While other methods used to predict clinical outcomes exploit the intensity of marker expression or cell proportions (such as [21, 62]), PH measures

characteristics of the shape of the point cloud in high-dimensional spaces that are not accessible with other methods. These topological features account for variations in the data due to noise or calibration of the flow cytometer. However, several variables considered during routine, manual inspections performed in the clinic (such as the range of expression levels or the mean fluorescence intensity) can be affected by data transformations. Fully automated and validated, this methodology could be readily integrated into clinical practice. Based on our observations, we propose three practical approaches to assess the risk of relapse on diagnosis, depending on the available methodology (see Fig 5). Each method requires biomarkers CD10, CD20, CD38 and CD45 to be included in the FC data.

1. When only visual inspection of 2D projections of FC data is possible, i.e. when neither PH nor ML methods are available, insights from PH applied to 2D projected data may help to identify qualitative patterns that distinguish R and NR patients. As seen in the Table from Fig 5, patients at risk of relapse tend to have large enclosed voids in most pairwise combinations of the biomarkers and smaller clusters that are far apart from each other in the CD10-CD20 projection. The NR group also exhibit smaller and larger loops in some projections including CD38.

2. If PH methods are available but not ML, then PH analysis should be applied to all 2D projections of biomarkers CD10, CD20, CD38 and CD45 and the numbers of long bars in the dimension 0 and dimension 1 barcodes computed. If, for the dimension 1 barcode, the distance of the corresponding PT curve to the mean PT curve of NR patients is larger than the distance to the PT curve of R patients, then the data is likely to correspond to a relapsing patient. The same analysis can also be applied to the CD10-CD20 projection in dimension 0.

3. If both PH and ML analyses are available, we suggest PH analysis of the 4D datasets, followed by transformation of the barcodes to PIs. Using a classification algorithm on PIs, such as SVM, would allow a high degree of accuracy.

While our analysis reveals that TDA has great potential for assessing the risk of relapse on diagnosis, our study has several limitations. The size of the datasets used may render the results sensitive to overfitting. Despite the robustness to noise of topological methods, further validation of the method on different datasets is needed. While the numbers of NR and R patients were balanced with oversampling, this difference should be reduced in future work. Additionally, a larger study using bone marrow samples with a larger number of common, standardized IPT markers could be insightful [63], particularly when data would be collected using the same hospital protocols for preprocessing. The use of a single flow cytometer would further improve the accuracy of the study, as machines can differ with respect to the data scales and compensation techniques that they use. Another limitation of our study is the high computational cost of PH: we reduced our data to $10^4$ data points to enable computations in dimensions 0 and 1, and to $10^3$ data points to enable computations in dimension 2. This represented 10% and 1%, respectively, of the $10^5$ points selected from the whole bone marrow. Even when a representative subsample was selected using the max-min algorithm (see 'Methods'), it took on average a week to compute all 16 pairwise combinations of IPT markers, with high-performance computing machines (see 'Methods' for details). However, as PH is an active research area, these computational limitations are constantly being improved. Finally, PH does not consider information about the intensity of markers or cell proportions. It would be interesting to incorporate such information into future PH analysis.

An extension of the methods presented here could be to apply our techniques to other FC datasets, monitoring patient responses to e.g. chemotherapy, transplant or CAR-T cell infusion. This could reveal new information about the emergence of resistance to these treatments.

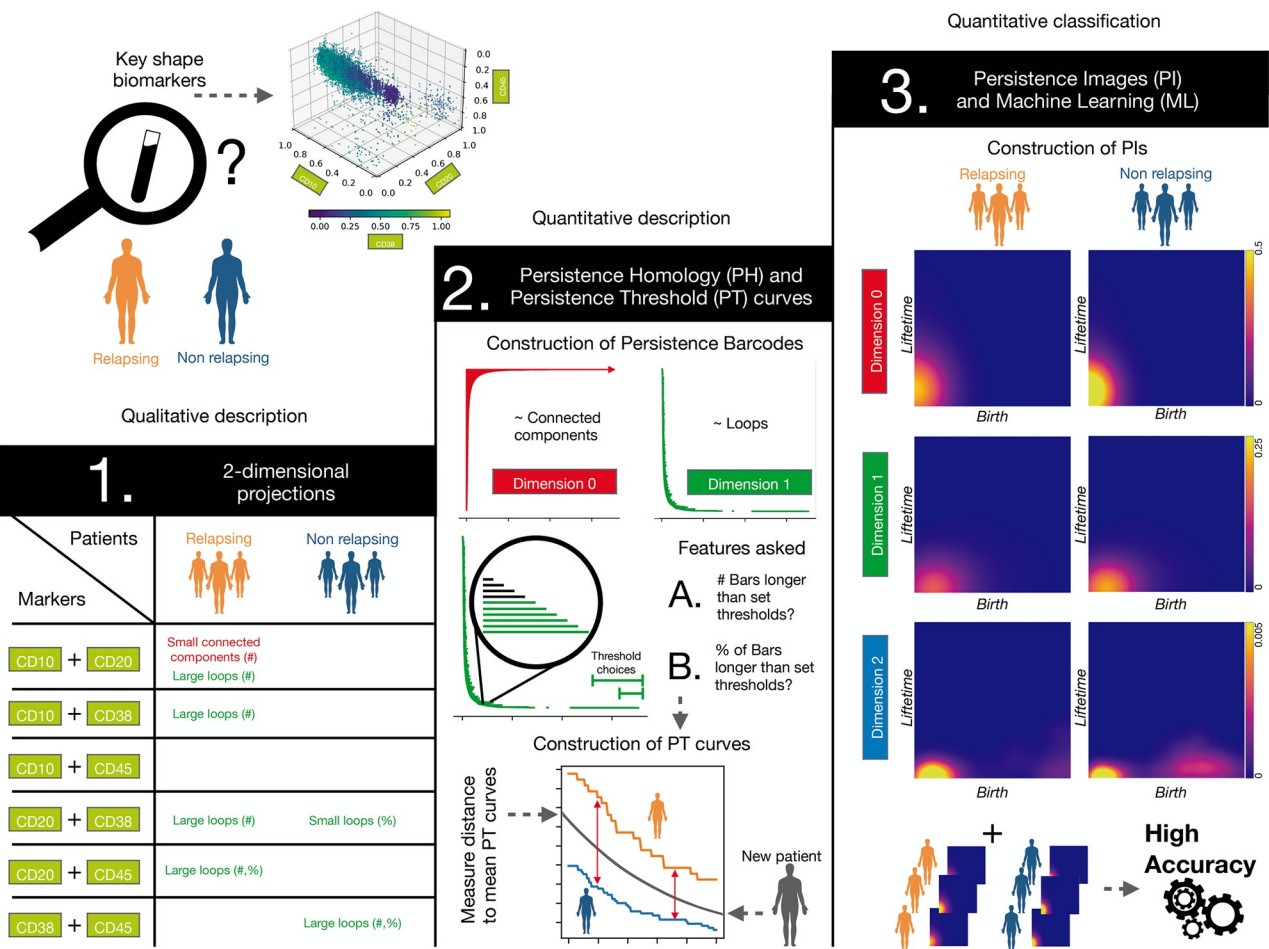

**Fig 5. Summary of results and methodology used.** We consider biomarkers CD10, CD20, CD38 and CD45 to compare FC data from R and NR patients. **1.** Two-dimensional projections of these biomarkers show differences between both cohorts in the number (#) and percentage (%) of both connected components and loops in the data point clouds. **2.** If PH is available, PH barcodes should be computed for each pairwise combination of biomarkers in dimensions 0 and 1. One can construct Persistence Threshold (PT) curves for each combination and then quantify differences between NR and R patients according to A.) the number (#) and B.) percentage (%) of connected components and loops. For a new patient, we can measure the distances of their PT curves to the mean PT curves of NR and R patients associated with each combination of biomarkers. We can then predict the risk of relapse by estimating an averaged probability based on the distances to the corresponding PT curves. **3.** For classification, Persistence Images (PIs) can be constructed from the persistence barcodes. Given the results in Table 1, this could lead to accurate classification of R and NR patients via Machine Learning (ML) with methods such as Support Vector Machines.

FC data can also provide information about all bone marrow cells, not just B-lymphocytes, and is therefore routinely collected for many haematological conditions such as lymphomas, T-cell leukaemias, myeloblastic disorders. Our analysis could be adapted to study these diseases. Combining information from IPT cell markers with qualitative biomolecular or morphology features has been used in [64, 65] to perform risk stratification and ALL diagnosis. By combining such qualitative data with PH analyses of FC data it may be possible to further increase the accuracy of patient prognosis and relapse prediction, and the selection of effective personalised treatment strategies.

Our results could have been obtained using either PH or ML. However, their combination provides biological interpretability and also exploits the high-dimensional character of FC data, whose shape is currently manually assessed in the clinic by focussing on 2D projections

of this routinely-collected data. The approaches outlined in this article represent a first step towards a more general approach in which topological descriptors computed from FC data inform risk prognosis with high accuracy and interpretable results. In other datasets, the same systematic approach could be used to identify informative biomarkers. We propose our approach to be tested prospectively in clinic to determine its practical use. Finally, our work could stimulate further studies which investigate the potential of methods from topological data analysis to reveal new features of cancer relapse.

## Materials and methods

### Ethics statement

This retrospective study was designed in accordance with the Declaration of Helsinki, under the LLAMAT Project protocol (2018). Approval was granted by the institutional review boards (IRB) of the four participating local institutions: Hospital Universitario Virgen del Rocío de Sevilla, Hospital Universitario de Jerez de la Frontera, Hospital Clínico Universitario Virgen de la Arrixaca de Murcia, Hospital Infantil Universitario Niño Jesús de Madrid. Patients and/or their legal guardians signed a written, formal informed consent to participate in the study. Personal information was anonymised.

### Patients

The inclusion criteria for the study were B-ALL diagnosis between February 2009 and July 2019, age less than 19 years, and availability of bone marrow FC data in Flow Cytometry Standard 3.0 (FCS 3.0). A total of 129 patients were included in the study: 2 from Jerez Hospital (HJ), 54 from Virgen de la Arrixaca Hospital (HVA), 27 from Niño Jesús Hospital (HNJ) and 45 from Virgen del Rocío Hospital (HVR). After preprocessing and marker selection, 96 patients were retained for further analysis. A discovery set was constructed with 30 patients from HVR (dataset 1) and 20 patients from HNJ (dataset 2). A validation set was later reanalysed with 46 patients from HVA (dataset 3). In all datasets, we counted with 80 non-relapsing (NR) patients, with at least three years of follow-up with no evidence of minimal residual disease, and 16 relapsing (R) patients. Patients' characteristics can be found in Table A in S1 Text and S4 Data).

### Flow cytometry markers and data preprocessing

Marker expression was obtained on FACSCanto II flow cytometers, in accordance with the manufacturer's specifications for sample preparation. FCS files were first imported into FlowJo (Becton Dickinson, 10.6.1) and FACSDiva (Becton Dickinson, 8.0.1) and inspected manually by standard methods (see Fig D in S1 Text): Acquisition anomalies, margin events (measurements that match the maximum or minimum values for any parameter), doublets (cells that are analysed together, as a single event), debris and dead cells were removed [12]. Files were further processed in R (4.1) using FC specific packages from Bioconductor. Specifically, files were compensated and transformed with Logicle transform [66].

For each patient, we subsampled aliquots to the same number of cells and then brought them into a single file via nearest neighbour imputation [67, 68]. This resulted in one file per patient. We subsampled every patient file to $10^5$ cells. Batch effects were accounted for with a rescaling transformation. While PH analyses are robust to small perturbations in the data point cloud [27], they may not be robust to outliers. To ensure that outliers do not introduce spurious topological features, we chose the 0.001 and 0.999 quantiles to transform every intensity value $x$ to $x'$: $x' = \frac{x - x_{q0.001}}{x_{q0.999} - x_{q0.001}}$. This was performed to avoid outliers, where $x_{q0.001}$ is the

quantile 0.001 and $x_{q0.999}$ is the quantile 0.999. Subsequently, the common B-cell antigen CD19 was used to select the B-cell subpopulation.

Finally, we selected $10^4$ landmarks from the lymphocyte cloud by applying the max-min algorithm [69] to each group of markers considered. Because of computational constraints in persistent homology, we reduced the number of landmarks to $10^3$ for studies performed in dimension 2. The last step involved the selection of a common set of potential biomarkers for each patient; these included FSC-A, SSC-A, CD10, CD13, CD19, CD20, CD22, CD3, CD33, CD34, CD38, CD45, CD58, CD66c, IGM, cyCD3, cyMPO, cyTDT. We omitted geometric markers FSC-A and SSC-A from the persistence analysis, to prevent unconscious bias. Patients that did not contain the full set were excluded from the analysis.

## TDA and Persistence Images

*Persistent homology* (PH) is the most commonly used method from TDA. The technique provides information on topological features such as connected components and loops created from point cloud data. To apply PH, one needs to construct a filtration of simplicial complexes from the point cloud data, i.e. a sequence of nested graph-like structures that include nodes (e.g., the data points) and edges as well as higher-order connections (simplices) such as triangles and tetrahedra. A method to obtain such simplicial complexes is the Vietoris-Rips filtration which we use here: Given a point cloud such as the one shown in Fig 1E, we place a ball of radius $r$ centered at each point. We now connect two points by an edge, if the balls surrounding them intersect. If three points have pairwise intersecting balls, we consider the edges to form a filled-in triangle, if four such points have pairwise intersections, their edges form a filled-in tetrahedron. As the radius $r$ increases, the simplicial complex built on the data contains more edges, triangles, and tetrahedra until at some point all points are connected to each other forming a high-dimensional simplex. PH monitors how topological invariants such as connected components, loops and 3D voids are born and later die along this filtration as the radius $r$ grows. For example, initially for very small $r$ all data points form their own connected components. It is only as $r$ grows that components merge together, i.e. one component dies while the other lives on now including two data points connected by an edge. For each radius $r$, the homology group of the simplicial complex captures the connected components (dimension 0), loops (dimension 1), and voids (dimension 2).

The term 'persistence', or lifetime, refers to how long these topological features live within the filtration. This can be represented by *persistence barcodes*, which are collections of intervals $[r_b, r_d)$ where each interval represents the lifetime of one topological feature in the filtration. Intuitively, $r_b$ is the radius at which the feature is first observed in the filtration and $r_d$ corresponds to the radius when the feature dies in the filtration. For an example of a barcode, see Fig 1F.

To further quantify these features, here we constructed so-called *persistence threshold (PT) curves*: we counted the number and percentage of bars in each barcode whose corresponding topological features persist longer than a threshold taken from a range of thresholds. Specifically, we studied threshold ranges $\tau \in [0.03, 0.07]$ and $\tau \in [0.1, 0.18]$ for dimension 0, as well as threshold ranges $\tau \in [0.007, 0.015]$ and $\tau \in [0.04, 0.05]$ for dimension 1. We chose these threshold ranges to represent different scales of persistence ('medium' versus 'long' scales).

In our study, we further applied *Persistence Images* (PIs), which we generated from persistence barcodes. These images are real-valued matrices that can be used as an input into a variety of ML approaches and are then useful for classification. The construction of PIs depends on hyperparameters, such as the grid size (5x5, 10x10, 25x25, 50x50, or 100x100 in this study) and the spread of the Gaussian distribution (0.01 and 0.05 in this study) which is used when

transforming the information from the barcode to the image. A more detailed summary of TDA can be found in the S1 Text, as well as in further references [26, 54, 70].

## Machine Learning methods

Artificial intelligence methods, specifically ML, were used for two purposes: feature selection and classification. For the first task, we divided data into discovery and validation datasets as stated in Patients (see Table A in S1 Text). For feature selection and prediction, we used Random Forests [71], a popular and efficient algorithm based on model aggregation ideas. The number of estimators in Random Forests ranged between 20 and 100, with 10 iterations to account for robustness. Firstly, we computed receiver operating characteristic (ROC) curves for the classification of R versus NR patients. We used persistence barcodes of the pairwise combinations of the IPT markers to confirm their prognostic relevance. We extracted the following simple descriptors from each barcode: the maximum, minimum and mean persistence and the standard deviation in dimensions 0 and 1. We used the descriptors to create 8-dimensional topological feature vectors for each patient. IPT pairs with an area under the ROC curve of less than 50% were deemed to have very low information and excluded from subsequent analysis.

For our second task regarding classification, we applied Support Vector Machines (SVM) [72], as this method provides good decision surfaces by maximizing margins between two classes. In our case, we assigned R and NR labels to the data matrices obtained from the PIs of the patients. We used the same set division into discovery and validation for this method. We varied two parameters associated with the SVM classification kernel: $\gamma$, which represents the curvature of the decision boundary, and $C$, which is the trade-off between correct classification and the distance between the decision boundary and support vectors. We randomly selected the SVM hyperparameters by performing stratified k-fold cross-validation in the training set for a logarithmic range of values of the parameters $C \in [10^{-2}, 10^{13}]$ and $\gamma \in [10^{-9}, 10^3]$. We also used Logistic Regression to construct binary regressions. We computed the mean score of fitting the model after 6-fold cross-validation. R patients constituted approximately 20% of the total number of patients. In order to address this imbalance, we repeated the SVM method for classifying R and NR patients with oversampling on the first ones, i.e., duplicating PIs from the R group.

## Computing machines

The max-min algorithm and Persistence Analysis were run on six machines from the Oxford Mathematical Institute, each with 36 cores, with up to 3.9 Ghz speed, and 768 GB of RAM. Figures and other calculations were run on an iMac, running under Mac OS 10.15, with four i5 cores, 3.4 Ghz speed and 32 GB RAM.

## Software

PYTHON (3.1) was used for the computations. R (3.6.0) and RSTUDIO (1.2.1335) were used for data curation. Persistence barcodes and images were constructed using RIPSER (0.3.2) [73] and PERSIM packages (0.1.3) in the PYTHON SCIKIT-TDA toolbox. Note that one could achieve faster computations using the recently introduced RIPSER++ library [74]. FCS files were read using the PYTHON CYTOFLOW library (1.0) and FC libraries from Bioconductor (3.11) in RSTUDIO. FLOWJO (Becton Dickinson, 10.6.1) and FACSDIVA (Becton Dickinson, 8.0.1) software packages were used for manual gating of FCS data. Random Forest and Support Vector Machine methods were obtained from the PYTHON SCIKIT-LEARN (0.24.2) classification library. Clustering was performed using the FLOWSOM algorithm [60], normally used for FC data. This last is

included in the SPECTRE package (from R, v.0.5.0), embedded in PYTHON with the rpy2 module (v. 3.4.5).

## Supporting information

**S1 Text. Supplementary materials.** Content on Topological Data Analysis and Methodological structure. This last includes the data processing techniques, obtention of biomarkers, and the interpretation and classification on such biomarkers.
(PDF)

**S1 Data. Random Forest classification results for the discovery group.**
(CSV)

**S2 Data. Random Forest classification results for the whole dataset with stratified k-fold and oversampling.**
(CSV)

**S3 Data. Random Forest classification results for the validation group.**
(CSV)

**S4 Data. Patients' clinical information.**
(CSV)

## Acknowledgments

The authors thank Heather Harrington (Centre for Topological Data Analysis, Mathematical Institute, Oxford) for helpful discussions.

## Author Contributions

**Conceptualization:** Salvador Chulián, Bernadette J. Stolz, María Rosa, Víctor M. Pérez-García, Helen M. Byrne.

**Data curation:** Salvador Chulián, Álvaro Martínez-Rubio, Cristina Blázquez Goñi, Juan F. Rodríguez Gutiérrez, Teresa Caballero Velázquez, Águeda Molinos Quintana, Manuel Ramírez Orellana, Ana Castillo Robleda, José Luis Fuster Soler, Alfredo Minguela Puras, María V. Martínez Sánchez.

**Formal analysis:** Salvador Chulián.

**Funding acquisition:** María Rosa, Víctor M. Pérez-García, Helen M. Byrne.

**Investigation:** Salvador Chulián, Bernadette J. Stolz.

**Methodology:** Salvador Chulián, Bernadette J. Stolz.

**Project administration:** Cristina Blázquez Goñi, María Rosa, Víctor M. Pérez-García, Helen M. Byrne.

**Resources:** Teresa Caballero Velázquez, Águeda Molinos Quintana, Manuel Ramírez Orellana, Ana Castillo Robleda, José Luis Fuster Soler, Alfredo Minguela Puras, María V. Martínez Sánchez.

**Software:** Salvador Chulián, Álvaro Martínez-Rubio.

**Supervision:** María Rosa, Víctor M. Pérez-García, Helen M. Byrne.

**Validation:** Salvador Chulián, Cristina Blázquez Goñi, Juan F. Rodríguez Gutiérrez.

**Visualization:** Salvador Chulián, Álvaro Martínez-Rubio.

**Writing – original draft:** Salvador Chulián, Bernadette J. Stolz.

**Writing – review & editing:** Salvador Chulián, Bernadette J. Stolz, Álvaro Martínez-Rubio, Cristina Blázquez Goñi, Juan F. Rodríguez Gutiérrez, Teresa Caballero Velázquez, Águeda Molinos Quintana, Manuel Ramírez Orellana, Ana Castillo Robleda, José Luis Fuster Soler, Alfredo Minguela Puras, María V. Martínez Sánchez, María Rosa, Víctor M. Pérez-García, Helen M. Byrne.

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
