## [Decision Letter · Decision Letter 0]

4 Mar 2023

Dear Professor Byrne,

Thank you very much for submitting your manuscript "The shape of cancer relapse: Topological data analysis predicts recurrence in paediatric acute lymphoblastic leukaemia" for consideration at PLOS Computational Biology.

We appreciate your patience during the review process. As with all papers reviewed by the journal, your manuscript was reviewed by members of the editorial board and by several independent reviewers. In light of the reviews (below this email), we would like to invite the resubmission of a revised version that takes into account the reviewers' comments.

The reviewers are generally positive about the approach and its application in this work. However, they raise concerns regarding details of TDA, including the choice of ML model, and parameter settings. In addition, some comparison and discussion to other methods would strengthen the paper.

We cannot make any decision about publication until we have seen the revised manuscript and your response to the reviewers' comments. Your revised manuscript is also likely to be sent to reviewers for further evaluation.

Sincerely,

Stacey D. Finley

Academic Editor

PLOS Computational Biology

Douglas Lauffenburger

Section Editor

PLOS Computational Biology

Reviewer's Responses to Questions

**Comments to the Authors:**

Reviewer #1: This study proposes tools based on topological data analysis and machine learning to study flow cytometry data from acute lymphoblastic leukaemia. This high-dimensional data is known to help estimate the risk of relapse of the disease in children and adolescents. In particular, the authors try to identify features such as voids in flow cytometry data. They first carry out PH on pairwise projections of relevant biomarkers in the data, and extract a feature vector with summary information that they then use for random forest classification. Focusing in on biomarkers of interest, they then quantity the number of significant connected components and loops characteristic of relapse patients using persistence threshold curves and persistence images. Their analysis predicts relapse accurately based on this data and confirms biomarkers commonly used for relapse classification. Generally, the manuscript is well-written and the figures are compelling and summarize the results well. It could benefit from a more detailed discussion of insights on how to parameterize the various methods for this and other applications. I recommend the following revisions:

1. On page 8, the analysis for the 4-dimensional point clouds (before the discussion of persistence images) could be better motivated. Is the point of using the FlowSom algorithm simply to confirm the 0-dimensional PH results?

2. Is there any insight gained from the fact that there are significant differences between the discovery and validation sets for the 4-dimensional analysis of biomarkers? Since there are no differences based on the whole dataset, it seems like this would not be important to report on (say, in Supplement 2.3a).

3. What is the motivation for performing classification for the low or intermediate risk patients separately?

4. Choosing thresholds for persistence threshold curves in applications is always a challenge. It would be interesting if the authors commented on any insights gained on the appropriate choices for flow cytometry applications.

5. More generally, the discussion would be improved by a discussion of parameter choices (grid, coefficients, etc) for both PH (say, for the PI images) and for the ML methods used, such as SVM. In particular, does this work suggest ways to choose these parameters in an informed way?

Reviewer #2: Summary:

In this manuscript, the authors apply a combination of topological data analysis (TDA) and machine learning (ML) to predict relapse in patients with acute lymphoblastic leukemia (ALL). First, they compute persistent homology of 2D point clouds obtained from pairwise combinations of 16 immunophenotypic markers in flow cytometry data. A random forest model, trained on dim 0 and dim 1 topological features, is used to classify patients into two categories: relapsing (R) and non-relapsing (NR). The cross-validation accuracy of the model is used for feature selection, which successfully identified four biomarkers, namely CD10, CD20, CD38 and CD45, known to be predictive of ALL prognosis.

Next, the authors introduce a topological summary statistic, the persistence threshold (PT) curve, which measures Betti numbers at various persistence (interval length) thresholds. They discover statistically significant differences in the mean PT curves (computed using pairs of previously identified biomarkers) between the R and NR patient cohorts. In particular, the presence of large numbers of persistent connected components (clusters) and loops (empty space) in the CD10-CD20 point clouds of the R cohort are evidence of heterogeneity in the B-cell populations of these patients. This result highlights the usefulness of TDA as a tool for measuring the effectiveness of therapies targeting gene alterations, deregulation, etc. responsible for increased heterogeneity in this subpopulation.

Lastly, the authors apply TDA directly to 4D point clouds containing all four previously identified biomarkers. Although summary statistics, including PT curves, are unable to classify R and NR patients with high accuracy in this setting due to loss of information, persistence images achieve state-of-the-art (100%) accuracy using logistic regression and SVM classifiers.

Major Comments:

1. The following statement presents an exceptionally narrow and outdated view of ML:

“Data analysis methods, such as principal component analysis or machine learning (ML) methods (neural networks, support vector machines, etc) could also be used for classification. However, these methods focus on identifying linear relationships between the biomarkers and do not characterise the shape of the data. Further, the results are often difficult to interpret.”

Several methods for non-linear dimensionality reduction, manifold and representation learning are available for the analysis of biomedical data. Furthermore, there are multiple model-agnostic techniques for performing feature selection and interpretation at both global and local levels. Please see “Interpretable Machine Learning” by Christoph Molnar (available online) for more information.

2. How does the TDA-based approach compare against other supervised and unsupervised methods for patient classification, e.g. CellCnn (Arvaniti and Claassen, Nature Communications, 2017), Citrus (Bruggner et al, PNAS, 2014), FloReMi (Van Gassen et al, Cytometry A, 2016)?

3. Is it possible to achieve similar or better classification accuracy with lower computational cost by training the classifier directly on 2D projection coordinates (e.g. using kernel density estimation) or other low dimensional embeddings obtained using an autoencoder, Isomap, MDS, etc.?

4. Dimension 0 homology is sensitive to total number of points in the point cloud. Were equal number of samples used to compute PH for each patient? Were the PT curves and persistence images normalized?

5. The dataset consisted of 16 R (17%) and 80 NR (83%) patients. These are divided into discovery and validation groups. The discovery group (Datasets 1 and 2, Table S1) contains approx. 25% R and 75% NR patients from two hospitals, while the validation group (Dataset 3, Table S1) contains 7% R and 93% NR patients from another hospital. Why were the training and validation groups generated in this manner? Would a naive classifier that always predicts NR achieve 0.93 accuracy? Why not merge data from different hospitals and use stratified k-fold validation?

6. Is the accuracy of model predictions in Table 1 the mean values across 6 folds? Please include std. dev., precision, recall and F1 scores in your results. For unbalanced data, precision, recall and PRC curves are more informative than accuracy and AUC scores.

Minor Comments:

1. ‘Robustness to noise’ is cited as an inherent advantage of topological methods. While the stability theorem guarantees robustness to small perturbations in the point cloud, TDA is not robust to outliers. Outlier removal in data preprocessing is important in this regard. Please consider highlighting this in relevant methods section.

2. Each patient was subset to 10^5 cells. From these, 10^4 landmarks were selected. For dim 2 persistent homology, the number of landmarks was further reduced to 10^3. Did the authors consider using Ripser++ or weak alpha filtrations to speed up the computation?

Reviewer #3: This paper provides a significant contribution in the prediction of ALL patient relapse by using flow cytommetry data. The authors exemplify promising results using persistent homology to summarize the point cloud data arising from flow cytommetry measurements to yield topological features that can be used for accurate classification as well as to give insight and augment visual inspection of the data. I have two minor comments:

1. The authors used an oversampling method to balance the data when they want to perform classification using Logistic Regression and Support Vector Machines. However, when using Random Forests to choose the pairwise biomarkers, they used the original data set without oversampling. Could this create biased results in choosing the pairwise biomarkers?

2. On lines 488-489 the authors mention that they used a Random Forest algorithm due to its accuracy and interpretability, and then on lines 500-501 the authors mention that they used SVMs since they provide good decision surfaces. Can the authors comment on why they expect different results from a Random Forest model as opposed to an SVM? One would expect a Random Forest to perform similar to an SVM since it relies on bootstrap aggregation method, which has the consequence of smoothing the decision surface. (See Breiman 1996, Bagging Predictors, Machine learning 24, 123-140).

**Have the authors made all data and (if applicable) computational code underlying the findings in their manuscript fully available?**

Reviewer #1: Yes

Reviewer #2: Yes

Reviewer #3: Yes

PLOS authors have the option to publish the peer review history of their article (what does this mean?). If published, this will include your full peer review and any attached files.

Reviewer #1: No

Reviewer #2: No

Reviewer #3: No
---

## [Decision Letter · Decision Letter 1]

6 Jun 2023

Dear Professor Byrne,

Thank you very much for submitting your manuscript "The shape of cancer relapse: Topological data analysis predicts recurrence in paediatric acute lymphoblastic leukaemia" for consideration at PLOS Computational Biology. As with all papers reviewed by the journal, your manuscript was reviewed by members of the editorial board and by several independent reviewers. The reviewers appreciated the attention to an important topic. Based on the reviews, we are likely to accept this manuscript for publication, providing that you modify the manuscript according to the review recommendations.

The reviewers raise two important points that still remain to be addressed: comparison to other methods outside of topological data analysis and providing code that allows reader to reproduce the results.

Sincerely,

Stacey D. Finley, Ph.D.

Section Editor

PLOS Computational Biology

Douglas Lauffenburger

Section Editor

PLOS Computational Biology

Reviewer's Responses to Questions

**Comments to the Authors:**

Reviewer #1: The authors have addressed all my questions and suggestions.

Reviewer #2: I am generally satisfied with the revisions made by the authors. I recommend a minor revision:

1) Please specify the numbers of R/NR samples in Fig. S8(A) and in Tables S6-S8. For example, in Table S6, approx. 2 NR samples are being classified in each fold. The total number of samples is 77 (why?). Each fold will contain approx. (1/6)*77*0.1667 = 2 NR samples (80 R + 16 NR => 16.67% NR samples in each of the 6 stratified folds).

2) Correct the typo in Table S6 (precision = 75.00)

3) Update and refactor your GitHub code (see comments pertaining to PLOS Data policy)

4) I would prefer to see at least one comparison with a non-TDA technique. You can classify the 2D projections used in clinical practice (no dimensionality reduction) by directly training on the coordinates instead of the PH features. You could also consider training your classifier on other shape features (elongation, convexity, etc.) to show that TDA perform equally well if not better.

Reviewer #3: The authors have addressed the concerns mentioned in my previous review. I recommend to accept the paper for publication.

**Have the authors made all data and (if applicable) computational code underlying the findings in their manuscript fully available?**

Reviewer #1: Yes

Reviewer #2: **No: **Although the authors provide source code via a GitHub repository (https://github.com/salvadorchulian/shapecancerrelapse), it is not possible to reproduce their results because:

1) The code is provided with no documentation or comments. Basic code formatting guidelines are not followed.

2) The code does not work. For example, in Result1/__Step 01.py the line tube=flow.Tube(file=datafile) returns an error because flow was never defined or imported.

3) The repository was last updated 2 years ago (on Dec 21, 2021). The code is outdated and does not reproduce any of the revised results (precision, recall, etc. in Tables S6-S8)

Reviewer #3: Yes

PLOS authors have the option to publish the peer review history of their article (what does this mean?). If published, this will include your full peer review and any attached files.

Reviewer #1: No

Reviewer #2: No

Reviewer #3: No

Figure Files:

Data Requirements:

Reproducibility:

References:

---

## [Editor Report · Decision Letter 2]

5 Jul 2023

Dear Professor Byrne,

We are pleased to inform you that your manuscript 'The shape of cancer relapse: Topological data analysis predicts recurrence in paediatric acute lymphoblastic leukaemia' has been provisionally accepted for publication in PLOS Computational Biology.

Thank you for thoughtfully and fully addressing the previous comments.

Best regards,

Stacey D. Finley, Ph.D.

Section Editor

PLOS Computational Biology

---

## [Editor Report · Acceptance letter]

8 Aug 2023

PCOMPBIOL-D-22-01436R2 

The shape of cancer relapse: Topological data analysis predicts recurrence in paediatric acute lymphoblastic leukaemia

Dear Dr Byrne,

I am pleased to inform you that your manuscript has been formally accepted for publication in PLOS Computational Biology. Your manuscript is now with our production department and you will be notified of the publication date in due course.

With kind regards,

Timea Kemeri-Szekernyes
